# Eco Trends, Counseling and Applied Ecology in Community Using *Sophia*

**DOI:** 10.3390/ijerph18126572

**Published:** 2021-06-18

**Authors:** Vasile-Petru Hategan

**Affiliations:** 1Institute of Media and Social-Humanitarian Sciences, South Ural State University, 454080 Chelyabinsk, Russia; vphategan@gmail.com; 2Institute for Social and Political Research, West University of Timisoara, 300223 Timisoara, Romania

**Keywords:** ecophilosophy, ecotheology, applied ecology, environment, philosophical counseling, spiritual counseling, community, Sophia, innovation

## Abstract

The study investigates the current trends that manifest themselves in two areas that have common origins in antiquity, found in the Hellenistic concept of Sophia: present in philosophy, where it expresses the love of wisdom, but also in theology where it represents divine love. Looking at this approach, the Sophia has manifested various orientations, either toward the field of ecology and the environment through the emergence of new concepts, such as ecophilosophy and ecotheology, but also toward the practice applied to the person, through the philosophical counseling or spiritual or pastoral counseling. This paper analyzes the characteristics of the applied trends, ecophilosophy and ecotheology, through their comparative analysis, along with a bibliometric study on papers published on these topics in indexed databases in the last 45 years. The paper presents the openness to innovation, through the emergence of the two concepts analyzed which created methods and tools specific to philosophical or spiritual counseling, and adaptations of these practices to the needs of contemporary society. Therefore, the innovation is sustained by creating a new specialization in community counseling practice, called eco-counseling for community (EC4com), with the new ecological trend, which can be included in the philosophical and spiritual practices applied in communities through individual counseling, for groups or community.

## 1. Introduction

The paper identifies some trends manifested in philosophy and theology, both fields with origins in the antiquity of humanity, these being dual defined by the concept called Sophia. Sophia has since had many representations, one being “love of wisdom” in philosophy and “the divine wisdom” for theology. Sophia has been defined by the ancient sophists as a practice of wisdom, representing “the knowledge of divine and human things”, and Plato can be considered the theologian of the ancient world, being the first philosopher to refer to theology because “without him, no he would have kept from theology neither the name nor the matter as such” [1]. The meeting of the two fields was noted by the philosopher Pierre Hadot, who studied ancient philosophy and its relationship with theology. Hadot indicated that philosophy in the Middle Ages was considered a servant of theology, by assimilating many philosophers, who became “Christian philosophers” [2]. They brought into the Christian practices some elements of philosophical practice, respectively, the spiritual exercises, being taken as practices by both fields [3].

Starting from their common origins in antiquity, the fields we are referring to show, especially in the 20th Century, various orientations toward practice, each maintain the specificity of the field. The paper refers to two of these applied practice trends, to individual or group counseling practices, developed in the form of philosophical counseling or spiritual counseling, which have approaches to the field from which they come. We study the ecological trend and orientation toward the environment, manifested in the form of the concepts of ecophilosophy and ecotheology, with each having a new approach to the basic theoretical field from which it comes. All these approaches are shown in Figure 1.

In the theoretical section, we will refer to the emergence of counseling concepts and trends toward ecology and environment areas, to argue a need to apply them in communities, resulting from the interdisciplinary links of philosophy and theology. Through analyzing the literature published on this topic, we found that in the applied approach of the two, manifested in the form of an advisor, most works take the form of volumes published by philosophers or theologians, in individual or collective thematic volumes. For this topic, the number of papers indexed in international databases is insignificant for the period analyzed in the paper. Starting from this established impediment, we compare the main characteristics found for each type of counseling, in order to highlight a common origin as well as some differences in approach, which derive from the specifics of each work. Regarding the ecological and environmental approach, we found a greater presence of published papers which presented the two concepts, ecophilosophy and ecotheology, and which we will analyze with the bibliometric method to highlight conclusions about the need to include them in programs for the person or for the community.

In order to have these analyses and the proposed purpose, in the debate section we present the resulting arguments, to give an answer to the following questions:Q1:What are the characteristics of the practices applied in the form of counseling in the two areas analyzed?Q2:What are the current trends in each area regarding the use of ecological concepts and environmental protection?Q3:Can these trends be considered as conceptual innovations or adaptations of applied practice in the two areas analyzed, and if they can they be oriented to the benefit of communities?

This study is an exploratory one, due to the rarity of the previous literature. This leads us to the definition of innovation, manifested somewhat modestly, but can be seen in the context of introducing the characteristics of ecology and environmental protection in philosophy and theology, through the emergence of new concepts. Referring to eco-philosophy and ecotheology, we try to answer if they can be included in the practices of counseling, as specialized forms of philosophical counseling or spiritual counseling. We analyze these practices comparatively in the paper. The conclusions will outline some perspectives of collaboration of the two fields, through the tendencies manifested in the sphere of counseling, but also of the ecological trend applied in the common benefit of humanity, on various levels of the work, both at the individual and group or community level. The purpose of taking over these philosophical or spiritual practices is highlighted by each analyzed eco concept.

## 2. Theoretical Background

### 2.1. Nature and Ecology in Philosophy and Theology

The approach to the idea of nature has been analyzed in philosophy since antiquity, in the mechanistic form expressed in Greek cosmology by followers of Pythagoras and Aristotle, followed by Renaissance cosmology that combatted ancient currents and introduced materialist ideas about nature, which became new cosmologies of the age. Modern cosmology introduces the concept of evolutionary biology supported by modern physics, by promoting the finiteness of nature, presented by the British philosopher Collingwood. This also indicates a direction for the future, moving from the idea of nature to the idea of history, indirectly highlighting the first trends of collaboration between fields [4]. Looking from a spiritual perspective, we find this approach to be a defining pillar of the church, to which are added the spiritual practices developed over the centuries and which have led to the outline of theological positions manifested in the form of eco-spirituality or ecotheology [5]. There are some theologians who present nature as a creation of divinity for the benefit of man and for his sake. Others put man in the position of a mediator of the relationship between nature and God, and others are against attracting ecological concepts in theology, considering “the concerns that it takes man away from the spiritual life” [5]. Berdyaev was a Russian philosopher who argued that any suffering in the material plane also attracts a spiritual problem [6], an aspect that needs to be evaluated by a specialist, preferably in applied counseling, in our opinion.

The concept of realistic spiritualism is promoted as: “In the spirit takes place the meeting between man and God”, and here takes place the transition from metaphysical naturalism to contemporary existentialism [7]. In the same context, Berdyaev stated in the middle of the 20th century that “I am interested not very much in characterizing the environment as in characterizing my reactions to the environment” [8], a conclusion that can be considered an urge for philosophers to be concerned with the ecology field by analyzing the phenomenology of perception expressed by Merleau-Ponty [9], about the connection between consciousness and body and its connection with the world, the concept can be extended to ecophilosophy, if we look at the connection between man and nature as a relationship derived from ecology. In this context, a natural problem arises regarding technology, in the sense that its evolution will influence humanity in relation to the environment. A study on this topic presents two optional answers: one in which technology influences the relationship, in the sense that it removes man from nature because of using new technologies that bring with them the so-called virtual realities, through which he can know nature via a computer. The new type of communication will create a distance of man from the natural environment which can be considered damage to the relationship between man and nature.

The second option is the diametrically opposite approach, which refers to the influence of technological progress on these relationships, helping man to improve his perception. In this way man can overcome the form of the personal experience of knowing nature or eliminate situations in which man cannot enter directly to visualize a particular phenomenon or state of nature; however, with the help of technology man will be able to perceive and understand them, beyond the limits of direct perception, facilitated by the physical body. For this reason, the model of perception presented by Merleau-Ponty can be extended when we talk about ecophilosophy [10] being seen as an extension of philosophy, in an attempt to highlight the connection of man with the environment in which he lives, using the tools and concepts from philosophy. If Spinoza considered nature “an extension of God” [11], the British philosopher Francis Bacon argued the opposite, that nature exists to be used by humans, thinking of the introduction of technology as an effect of the development of science [12]. Furthermore, Descartes stated that nature is mastered by the humans [13]. Later, in the middle of the 20th century, Aldo Leopold claimed that humans are part of the planet’s biosphere, and they have responsibilities to it [14]. He was the first to refer to the concept of the ethics of the earth, when referring to the phrase “the land ethic”, which later became the concept of environmental ethics [15,16] and which is considered necessary in the evolution of humanity [12], and a true paradigm of the new concept of environmental ethics [12,17].

Recently, other researchers have questioned whether religious beliefs can influence environmental ethics, starting from the fact that religions have had some attitudes toward nature [18,19,20,21,22]. One of these was found in Christianity, in which man received the earth to master it, and the hierarchy becomes one like God-man-nature [12]. This approach was reached much earlier by Naess, who spoke about a concept called the deep ecology movement, which tends to have some characteristics similar to Christianity, and his attitude toward nature may involve the discussion of the Christian or Oriental religion [23], which implicitly generates man’s responsibility toward nature, through care for the environment, regarded as a gift of divinity. The movement generated by Naess starts from the philosophical approach to nature, which brings attention to an innovative concept, ecophilosophy, seen as an “ecological wisdom”, or an expression of philosophy in the field of nature [24,25]. Naess says that nature should not be controlled by humans in its own interest, and the concept of ecophilosophy is not an academic philosophical concept and should not be interpreted as an ideology or religion [26]. For this reason, he is inclined to create a pro-ecology movement that will generate some changes in the community, using the levers offered by the philosophical or spiritual concepts identified in his research.

Other researchers of the ecological trend have highlighted two directions, one being the one in which philosophers had direct experiences in nature, such as Naess, and which brought eco-concepts in support of humanity. Those who remained in the theoretical area of the academy also lacked experience, which is why they went to study the concepts of environmental ethics [27]. These fields can work successfully together, generating bio-cultural interference [28,29]. Philosophy is associated with the sciences of life, through other approaches [30,31], which highlight the applied characteristics of the newly introduced concept ecophilosophy [32,33], but also which have been analyzed comparatively to promote this practice in communities in the form of a program called ecophilosophy for community (EP4com) [34].

The concept of ecophilosophy has an inferior status to philosophy as a field, being similar to the genus–species relationship existing in biology [35]. This fact has been an adaptation of philosophy to current environmental requirements. By introducing the concepts of ecophilosophy, ecotheology and environmental ethics, it is desirable to change the attitude of man towards nature, from master to protector, who develops an ecological consciousness [36] as a form of social consciousness. In this context, man becomes part of nature, developing some values, such as cultural, ethical, aesthetic, or spiritual or religious, all of these expressing a symbol of his evolution [36]. This confirms the vision expressed by the philosopher Solovyov, for whom man is in the “center of the universal consciousness of nature” with an almost messianic role, as its savior, placing man in an eco-centric position as a culmination of divine creation. This appeals to Sophia as an expression of divine wisdom [36,37] through the concept of sophyology, promoted by him. Starting from Eastern religious beliefs, Mathews refers to redefining the meaning of life, with the help of self-realization [38], where environmentalism can be a way of life, based on the concept of ecophilosophy [12]. 

To show the connections between philosophy, theology and science, Solovyov refers to the concept of free theosophy as their free association, and the practical ideal calling it free theocracy, which will generate so-called free theurgists and express the connection between Sophia and Christianity [39]. The major challenge for both areas remains the maintenance and development of the dialogue between philosophy and theology, both facing some crises of authority manifested by contemporary society, where Christian philosophy can provide its own practices and tools that precede theology and complement spiritual therapies already offered by the counseling developed by both areas [40]. Ecology entered as a concern in Christian theology, as of the 1970s, in the form of positions, and in the care of the environment, a discourse that continued in the 1990s, being present in all denominational traditions and theology schools. The task of Christian ecotheology has become a major one, to develop a liturgical vision of the world [41]. Books and articles on the challenges of ecology to environmental ethics have been published since the 1980s, referring to theology or the connections to it through the use of the hybrid concept of ecotheology [42,43,44,45,46,47,48,49,50].

1990 represented a moment of impetus given to this trend, through the open letter launched at the Moscow Global Forum of Spiritual and Parliamentary Leaders Conference, where 291 spiritual leaders addressed religious communities and scientists to work together to protect the environment [51]. Through the importance of the concept of wisdom and the fact that the sciences are not enough to solve environmental problems, theology was tempted to develop the foundations of a relationship between deity and humanity, appealing to this hybrid concept. Thus, the ecotheology focus on derived concerns in ecology, such as Christology, Theodicy, Spirit, Eco-Feminist Theology or Eschatology [52].

The role of theology became an active one, through concerns related to the environment but also for developing a better relationship of humanity with it. Later, the ecumenical partners organized themselves into eco-theological networks, integrated into the ecumenical movement, starting from the fact that eco-theologists have great potential to improve the needs of the community and can be oriented towards the development of good practices in the field of environmental protection [53]. The authors of this idea support the realization of an eco-theological consciousness, which implies respect and care for the community, and which can be developed with the help of eco-theological methods that can be implemented through spiritual practices [54]. Through all these approaches, we affirm that the new concept of ecotheology represents the ecological vision of theology, which addresses the inhabitants of the planet, through theology, as an important vector of the spirituality of humanity.

We notice that in the theoretical debates that existed in the analyzed fields, philosophy and theology, there is a third field represented by ecology, which facilitated their innovation by generating two hybrid concepts, and interfering with both, according to Figure 2. Regarding community, we refer to all its components, to individual persons, to groups of persons, to organizations and institutions that operate in the community and to the community as a whole. Outlined on this structure are the specializations for counseling, which are represented in the following Figure, and which together give a whole, called the community.

Figure 2 shows the apparition of innovative concepts, ecophilosophy and ecotheology, each having incorporated an ecological side, specific to the current trend of environmental protection, which folds in the theoretical content of each field. All these occur in an attempt to become applied, by training tools for each field, towards an ecological approach as a final goal. If initially this innovation process consisted of clarifying the concepts developed by interacting with the fields represented in Figure 2, the next step can be adaptation of the practices, and we will refer to the practice of counseling in both areas.

### 2.2. Counseling, as a Form of Practice in the Two Areas

We approach this topic to highlight the interdisciplinary connections of philosophical counseling practices, but also to combat the expressed opinion that it cannot occupy a distinct position in counseling, with the motivation that it has not reached a certain “degree of spiritual maturity”, and this opinion are reasoned by the fact that it does not following diagnostic procedures on the subject, like some therapies, and the appearance of the new practice shows the social involvement of philosophy, named appreciative philosophical counseling [55]. 

Looking in this context, man as a being anchored in reality needs more practical tools that can express the theory in a lived reality [56]. If some authors were only interested in assimilating spirituality in therapeutic practice this paper presents the links of philosophy in this field, through philosophical practices that have been rediscovered recently, having a tendency to define themselves distinctly from the practice of a philosopher, and using a new specialization of philosophical counseling, which can also have approaches in the field of personal spirituality.

In Europe, philosophical counseling, as a practical philosophical form, was first promoted by the philosopher Gerd Achenbach, who in 1981 announced the opening of a philosophical practice office in Germany with the first clients who were assisted by the practitioner in a process of counseling in which he used the knowledge of the philosopher [57]. He is considered in Europe to be the “father of philosophical counseling”, a title later disputed with practitioners on the North American continent [58]. Achenbach remains a successful reference on the global trend of launching a philosophical practice, expressed in the form of specialization (philosophical counseling) unanimously recognized and promoted by other practitioners. He, being a follower of philosophical practice adapted to the person, using tools from philosophy.

The practitioner will support the person in clarifying life situations and in developing thinking skills based on concepts from philosophy, which have an active role in the development of the person, in the form of a process of identification and construction of their own vision of life. The process was developed by other practitioners, one being Ran Lahav who gave up the chair to devote himself to contemplative philosophical practices. He highlighted the concept of the philosophical perimeter of the person, starting from the allegory of the cave described by Plato, in antiquity [59].

To understand the potential offered by the interdisciplinary approaches that can manifest in the field of counseling, we start from an analysis of counseling, which presents the counseling process as being based on communication and relationships generated by meeting other people. The elements of counseling are detailed as a communication between the practitioner and counselor or client; the process does not expressly seek to help the person, but rather generates a certain empathy that appears as a result of the meeting.

The process can generate a state of acceptance on the analyzed topic, by clarifying the subject. The practitioner uses the art of questions, to create a dialogue, based on evaluation, interpretation, investigation, support, consolation and clarification; seeking to support the counselee in creating his or her own vision of the world and life [60]. American theologian Caputo studied some characteristics of theologians and philosophers, which he said are attracted to the person’s life through counseling, but in which both positions exceed the limits through their interaction, inspiring each other, which is why in his conception they become religious philosophers or philosophical theologians [61].

Another theologian, Koestembaum, who became an organizational philosophical advisor, writes of philosophy as a secular, doctrineless priesthood that gives religion a justification and a vocabulary [62]. Pierre Hadot refers to the universities of the Middle Ages, stating that philosophy was seen as a servant of theology, while theology took from philosophy concepts such as logic or metaphysics, and spiritual exercises initiated in the schools of antiquity were already included in Christian spirituality. After the Middle Ages, where philosophy occurred only outside the university, at the end of the 18th century, it would return to universities, with the emergence of philosophers who restored the autonomy of philosophy from theology [63].

The key element that made philosophy monopolized by theology through inclusion in the Christian morality was by taking over its spiritual exercises, initially developed by ancient philosophical schools, which in the Middle Ages became an integral part of spirituality, specific for the monasticism of the times. The return to theory and philosophical concepts developed in the modern era, and made philosophy become “a way of life and a way of looking at the world, and it becomes a concrete attitude” [63]. To balance the interference of philosophy with other fields, we study the essential features of philosophical counseling which shows us the real stake, manifesting itself as a distinct specialization of the philosopher through the following facts: it is an open dialogue, and a sincere communication; the practitioner is paid and applies the special methods and the principles of ethics in his practice [64]. Counseling practices have generated debates about the role and involvement of the practice in the form of counseling or therapy [65], and was continued with studies on the needs to develop the skills for spiritual or existential counseling of the person [66], or by presenting new practice options, such as group facilitation [67], the use of contemplative or meditative applications [68,69] or implications and roles of the philosophical counseling in communication [70].

## 3. Research Methodology

In order to highlight the objective of the paper, qualitative research was carried out based on a comparative analysis of the main characteristics derived from the adaptation of practices to each analyzed field, philosophy and theology, manifested in the form of specialized counseling parts (philosophical counseling and spiritual counseling).

To highlight the differences that appear in practice, the two types of counseling, philosophical and spiritual, will be compared as practical forms dedicated to Sophia.

In order to study the biggest trends in the fields of ecotheology and ecophilosophy, as well as studying which authors have the most impact in this field, a bibliometric analysis was considered appropriate. The free software VOSviewer was used in order to identify the relationships between keywords used in literature and between authors and their citations. VOSviewer software (Centre for Science and Technology Studies, Leiden University, Leiden, The Netherlands) was chosen for data processing because it is an open source computer program for creating, viewing and exploring bibliometric maps and has previously been used in research in various fields to obtain relevant results [34].

At first, we searched on the Web of Science (WoS)—Clarivate Analytics database on the topic of “ecotheology” and “ecophilosophy”, in the years 1976–2020. The database evidenced 194 papers from all document types and all categories (Science, Social Science, Arts and Humanities). The list was saved as a txt file, and a thesaurus file was provided to merge almost identical terms. Afterwards, this file was processed through the VOSviewer software, providing an analysis of keywords and citations based on individual authors and countries.

## 4. Results

### 4.1. Analysis about the Philosophical Counseling Versus Spiritual Counseling

In the theoretical section we have made some references regarding the more recent appearance of some practices within the analyzed fields, usually manifested in the form of counseling, philosophical or pastoral or spiritual, which we will study further. These counseling practices applied to each field have been the subject of other research papers, being analyzed both individually [71,72,73] and in terms of the relationships between them [74,75,76]. The comparative study presented in Table 1 demonstrates that the philosophical counselor uses the tools of counseling, avoiding becoming a spiritual guide for the advised person [77]. In practice, similarities and differences have been identified between philosophical and spiritual counseling, which can be applied to the person and groups of people [78], with the help of a counseling process for each field. We will analyze two practices of counseling, philosophical and spiritual, in an attempt to identify many common elements that show the need for collaboration and mutual support, for the benefit of the counselor or counselee, and expressed in one of the forms of counseling discussed below.

In order to emphasize the originality of each practice, and the origin and adaptation to the practice of counseling in the field from which it comes, but also to confirm one of the research hypotheses (Q1), the characteristics of the forms of counseling were analyzed (Table 1). The results consist of identifying several common points, with the differences deriving from their way of working. The ways of approaching counseling are in accordance with the specialization of the practitioner and depending on the field in which his practice occurs. The interpretation of the characteristics shows an adaptation to the applied requirements that appear in each field studied, and the use of a counseling service as a form of philosophical or spiritual practice that supports the person, through specialized counseling offered by the practitioners from each field. This process can be considered an adaptation process, by the fact that the tools and methods used are taken from each field and are included in a counseling process, adapted to each philosophical or spiritual counselor [79].

We notice that both practices have the same recipients, being services that can be offered to them in the form of a specialized counselor, who is a trained specialist. He is free to choose the tools and methods used in his practice that can support the person or group in its evolution, with the help of applied counseling. The attachment of the pastoral counselor to a cult requires his permanent presence in the community, and his service is free; these are two aspects that can become major ad-vantages to be exploited in promoting the activity of pastoral/spiritual counseling. The analysis has certain limits, given the number of elements taken involved, but in this sense the paper can generate debates and new research topics for those interested in studying the proposed practices.

### 4.2. Bibliometric Analysis on the Concepts of Ecophilosophy and Ecotheology

The analysis of the papers showed that from the point of view of the types of documents, two thirds of the papers were articles in journals, followed by book reviews and book chapters. Regarding WoS categories, half of the papers were classified in Religion, followed by the categories of Environmental Studies, Ethics and Philosophy.

It can be noticed that in the first 25 years of the studied period only 18% of the papers were published. There has been a substantial increase since 2008. From the year 2011, the interest of the authors remained almost constant (Figure 3) with a maximum of 21 articles in the year 2018. In the last two years there has been a slight decrease in the number of articles, following the maximum in 2018.

Regarding the chronology of the two concepts, the first identified paper that addressed the topic of ecology was from 1976, and the field of ecophilosophy has been around since 1980. It should be noted that neither of the two articles had any subsequent citations.

The analysis presented in Figure 3 is limited to the last 10 years, as this period is the most representative of the evolution of the interest of publishing and presenting the concepts analyzed in the paper. The rest of the period had minor manifestations, conclusion given by the small number of publications that had some relevant approaches. We observe from the presented graph a trend oriented towards the increase of the number of works, with minor decreases in the last two years, due possibly to the effect of the COVID-19 pandemic.

#### 4.2.1. Analysis by Keywords

In this section, the analysis by keywords is made, presenting how the most frequent keywords appear together in the articles studied. The VOSviewer software evidenced 497 keywords. Some of them were very similar; therefore, a thesaurus was used in the analysis, replacing keywords such as “environmentalism” with “environment”. This gave a total of 490 keywords, and by setting the minimum rate of occurrence at three, the software found 35 keywords.

Their relationship is evidenced in Figure 4. The Figure created in VOSviewer has the following structure: larger dots represent keywords mentioned more often, thicker lines represent how often the keywords are found together, and the distance between dots represents the strength of the relationship between keywords. The colors represent the clusters identified in the analysis by VOSviewer.

In our data, by choosing a minimum rate of three occurrences of keywords and keeping the default resolution of 1.00, three clusters were found. These clusters are detailed in Table 2 and also in Figure 4, by the colors red, green and blue.

The top keywords in the articles analyzed are “ecotheology” (with 49 occurrences), “ecophilosophy” (32 occurrences) and “ecology” (20 occurrences). Other important keywords by occurrence (more than 10) are “environment”, “nature” and “sustainable development”. Of interest are also “religion”, “creation” and “earth”.

The keywords introduced in the analysis identified the links between them, as represented in Figure 4, to highlight their grouping into four clusters, marked by colors and defining each field to which they belong, such as: ecophilosophy, ecotheology and ecology. This confirms the hypothesis that the three fields can work together to achieve and develop an ecological trend, with major concerns for environmental protection, which is highlighted in the ecological cluster as having relations with the two concepts analyzed. From the representation of Figure 4 we observe the location of the concepts, where the ecological and environmental domain is positioned between the other domains: philosophy (focused on ecophilosophy) and theology (focused on ecotheology); thus validating some of the research hypotheses of this paper (Q1 and Q2).

The central cluster, colored green, relates the field of ecology with the environment, and its essential idea of studying earth science.

The biggest cluster (colored red) is related to ecotheology, and it contains religion and theology, being the main keywords in this cluster. The next cluster, colored blue, has its main concept “ecophilosophy” and suggests the link between sustainable development, climate change and environmental ethics, as a distinct branch of philosophy.

The bibliometric analysis of the analyzed concepts validates, through the three resulting clusters, the hypothesis in which the three domains may interfere by including the eco trend, generating the hybrid concepts marked ecophilosophy and ecotheology, and presented in Figure 4 and Table 2, in the clusters 1 and 2.

#### 4.2.2. Analysis by Authors and Countries

The papers analyzed were written by 201 authors belonging to research organizations from 30 countries, and according to the number of papers the first five places were US (28%), Canada (8%), Poland (7%), South Africa (7%) and England (5%).

Analyzing the affiliation of researchers who have published papers, it was found that there is an interest of authors in countries from all continents. In the North American continent or in English-speaking countries, more than half of the papers analyzed were registered, followed by authors speaking different languages from countries from Europe (a quarter of the papers), Asia (10%) and South America (6%).

From the distribution of the papers by the authors, 96% of them published a single paper in the analyzed fields. Given this situation, it is found that there were no links of collaboration of authors from different countries, and for this reason a graphical representation was not included.

#### 4.2.3. Citations by Authors Analysis

This analysis presents the most cited authors on the topics of ecotheology and ecophilosophy that are indexed in the WoS database. Using the threshold settings of a minimum of one document per author (which was the default) and at least five citations, VOSviewer evidenced 39 authors that met these conditions. Table 3 presents the authors with at least 10 citations, in descending order of the number of citations.

The results from the analysis showed that the most cited authors are Latour [80] with 45 citations and one document and Harrison [81] with 41 citations and one document. Other authors who have been positioned at the top, judging by the analysis, are Whitney [82] with 37 citations and two documents and Fairweather [83] with 33 citations and one document. In the medium position are the following authors: Fox [84] with 13 citations about the ecophilosophy topic and Bratton [85] with 11 citations about the ecotheology topic. The author Riley [86] registered the highest average citations of 9.62 from three documents. It can also be seen that the most cited authors have written papers in all fields, which shows that there is a high research interest.

## 5. Discussion and Future Directions

In the theoretical section we identified for the two areas analyzed a new conceptual trend, manifested by two concepts derived from the interference of each field with the third field: ecology. Ecology generated the trend called eco, which joins the basic concepts, resulting in two hybrid concepts called ecophilosophy and ecotheology. We consider this process a necessary one, open to innovation, which joins elements of the ecological field in the new conceptual framework. It includes these elements, for the configuration of a hybrid type concept, which also results from their name.

The bibliometric analysis was performed for the two hybrid concepts, and shows the interdisciplinary links that are made between the three areas analyzed, where ecology is positioned between them, influencing each basic area of analysis, namely philosophy and theology. We thus obtain a validation of the presence of the ecological trend within the analyzed theoretical domains (Figure 2). From a point of view, we analyzed counseling, as a common element of the two areas, to show their common interest in serving the person or community.

The practices we refer to are forms of manifestation in the form of a specialized counselor to each field, and we believe that they can support the needs of modern society by solving or clarifying them. The area of these needs is a dynamic one, starting from personal needs and continuing with group or community needs, and counseling can become an alternative in clarifying or solving problems, dilemmas or life situations, which the recipients of the practice can have. 

The proposed ecological trend is one that develops this area, by the fact that practical ecological counseling can solve situations with an impact on nature and the environment, with major effects on climate change, and thus these practices become necessary and useful to a community interested in this topic.

From the comparative analysis of the two types of counseling and their characteristics (Table 1), there resulted some similarities and essential differences, which individualize the counseling and are based on the tools, concepts and methods to each area. The tools and techniques used by practitioners in the two types of counseling are: personal confession, active listening and asking questions to clarify a dilemma or life situation, conducting a dialogue that facilitates personal or group communication, and using practices and exercises meditative or contemplative that can be used in solving the existential or life problems.

With the emergence of the proposal for the application of ecophilosophy in communities, in the form of the community approach initiated under the name ecophilosophy for community (EP4com) [34] we can consider this approach to be only a preamble for other community initiatives. For this reason, we extend that community approach to counseling, as a form of philosophical or spiritual practice, by taking over the two eco concepts of the hybrid types in community counseling. This approach will once again confirm the ecological trend we are referring to but also the opening of the two areas to practical approaches, manifested in the form of counseling, which can be brought to the benefit of the community through eco-counseling for community (EC4com), the proposed process of which is shown in Figure 5.

We consider it opportune and necessary to extend the community approach from ecophilosophy to eco-counseling, in which we also included spiritual counseling, which will take elements from ecotheology, bringing them all into the practice of a counseling, now a destination for community.

The active participants in this approach will be the two forms of counseling, joined by the concepts of ecophilosophy and ecotheology presented in the theoretical section, which can connect together to create a new program applied to communities, represented in Figure 6.

This hybrid approach, which combines two concepts and practices related to their basic fields, can be considered an innovative process in which the practice of counseling (philosophical/spiritual) puts into practice the ecological trend thus manifested for the benefit of the community. The approach we refer to can be achieved by initiating specialization programs for practitioners in philosophical and spiritual counseling, to develop new skills and abilities for counseling that consider the assimilated ecological element, in the form of eco-counseling for community (EC4com). The new program can be organized by the professional associations or universities, according to the pandemic situation, and the online version can be a new option for all participants, for safety [87,88]. When speaking about community, we refer to all its components, respectively, to individual persons, to groups of persons, to organizations and institutions and to the community as a whole. Already outlined on this structure are the specializations to counseling, which are represented in Figure 7 and which together give a whole, called the community.

We observe in Figure 7 who are the beneficiaries of the practice of counseling, and that can be a reason to reconsider their needs, solved by using techniques and methods of applied counseling, and using the work tools characteristic of the specialization and field. The approach of the topic was made from the position of research on the practices of counseling, and choosing spiritual and philosophical counseling. This was made to identify the conceptual links in the basic fields from which it derives, respectively theology and philosophy, and to study the options to interfere with the other field, that of ecology and environmental protection. The purpose of this approach is to present the way of adapting the respective practices to the counseling applied to the person, with the identification of a new trend in contemporary society. The paper used a comparative analysis of the main characteristics identified for the mentioned practices, spiritual counseling and philosophical counseling, in order to identify differences or common approaches, and to find their potential for adaptation and innovation by attracting new concepts derived from the eco-trend presented. Starting from the presented concepts, we also included their bibliometric analysis, the results being interpreted to show the existing connections between the basic domains, as well as the relatively low publishing options manifested for the analyzed domains. All these studies and comparative analyses have led us to the idea of developing the analyzed practices, which may include the ecological trend in a process of specialization of practitioners. This is a trend that can be considered the premise of a process of adaptation to contemporary society, through eco-counseling, the practice proposed in the paper. Returning to the practice of counseling, presented in the paper (Figure 7) were the beneficiaries of this service, from which we can expand its scope, by promoting eco-counseling in communities, a process that involves the option of applying it to people, groups or other entities from a community.

We consider the approached topic just a preliminary one, which can open new directions of approach and research of the concepts and practices analyzed now, by developing a program that can be implemented in the process of training and continuous training of counseling practitioners. Thus, we believe the specialization proposed by eco-counseling for community (EC4com) can become a useful program for communities, which can be monitored during its implementation and evaluated at the end, in order to improve it, or simply to report the effects and benefits resulting from this new practice. The ecological field that interferes with the two humanistic fields presupposes pragmatic approaches, determined by previous research in the field. It is for this reason we conducted this research starting from finding an interest in the ecological field and from the research and conceptual studies conducted in this direction, in the social sciences and humanities areas.

## 6. Conclusions

We can conclude that the two fields used innovation by creating the two hybrid concepts and bringing attention to the third field, ecology, by offering the concepts called ecophilosophy and ecotheology. Regarding the practical approach, the two areas resorted to counseling, through which they each brought in these tools and methods, which were adapted in the form of practices through philosophical counseling and spiritual counseling. Continuing with these trends, we believe that the practice of counseling can call for innovation again, by creating and implementing in community practice new forms of counseling, based on the same characteristics of the hybrid type, which is why we support the introduction of the concepts analyzed in a new specialization called eco-counseling for community (EC4com) through the applications for the person, groups or community (Figure 8).

This representation confirms an innovative or adaptive process, identified depending on the area where it occurs: a first innovation process, by involvement in the theoretical and conceptual area of a third field, the ecological, and the generation of new concepts; followed by an adaptation process, referring to the practices developed by the two fields, manifested in the form of specialized counseling; and an innovation process that can take place in the community approach, which will bring together the practice of counseling with the ecological trend assumed by both fields.

The performed analysis has some limitations by choosing the papers analyzed from a single database: WoS. This can be further developed by accessing other databases.

It can be observed that the theologian has a major advantage over the philosopher practitioner, to communicate within a community, through his activity and his permanent presence in the community, which encourages the expansion of the ecological trend in the counseling process presented as a practice of the two fields. Through this new approach, this potential can be highlighted and brought closer to the beneficiary, respectively, the person, group or community, through the new proposed concept, called eco-counseling for community (EC4com).

The paper presents the first direction of action, by explaining the concepts and initiating an innovative program, which brings the ecological trend to the attention of other practices analyzed, in order to be promoted and debated by those involved in training specialists and counseling practitioners, especially those to whom I referred, as a first step in the implementation of the program proposed. The second step is to promote the EC4com program to its beneficiaries, from individuals to groups and the community, and to evaluate the results over time in order to improve it. In order to achieve a so-called ”awakening” of humanity, we propose the use of other concepts generated by Sophia, such as ecophilosophy and ecotheology, by activating practices that derive especially from the areas analyzed, which can successfully take over the ecological trend we support.

The implementation of the EC4com concept can be transposed into community development policies and strategies, through the counseling practices presented, which can also take over the ecological concepts, for the benefit of the community and implicitly the benefit of the people within them.

## Figures and Tables

**Figure 1 ijerph-18-06572-f001:**
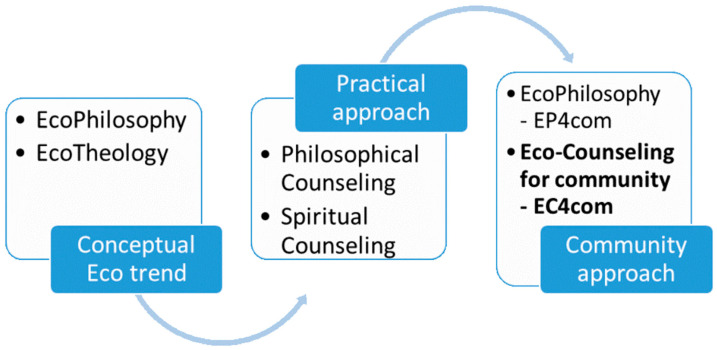
Connections between the trends developed by both areas analyzed.

**Figure 2 ijerph-18-06572-f002:**
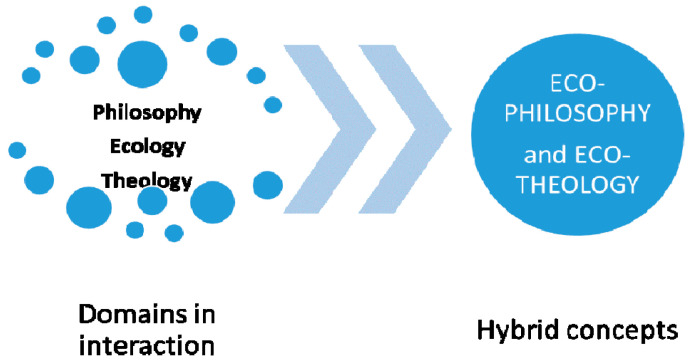
Interaction of three domains generates hybrid concepts.

**Figure 3 ijerph-18-06572-f003:**
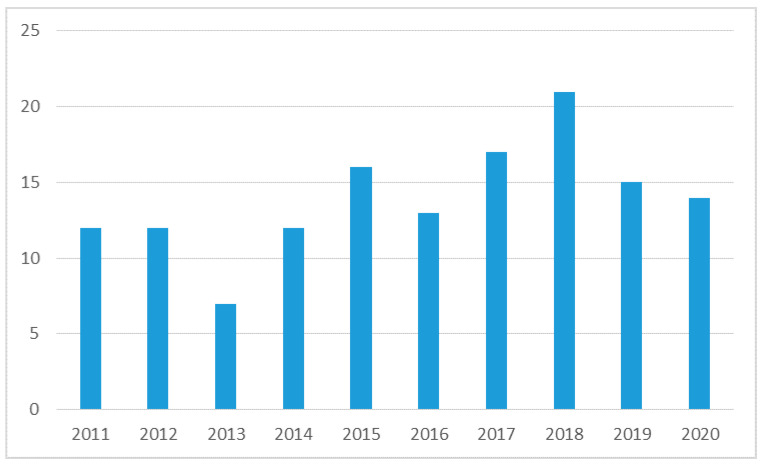
Number of papers published in the last 10 years. Source: Generated by the Web of Science analysis report.

**Figure 4 ijerph-18-06572-f004:**
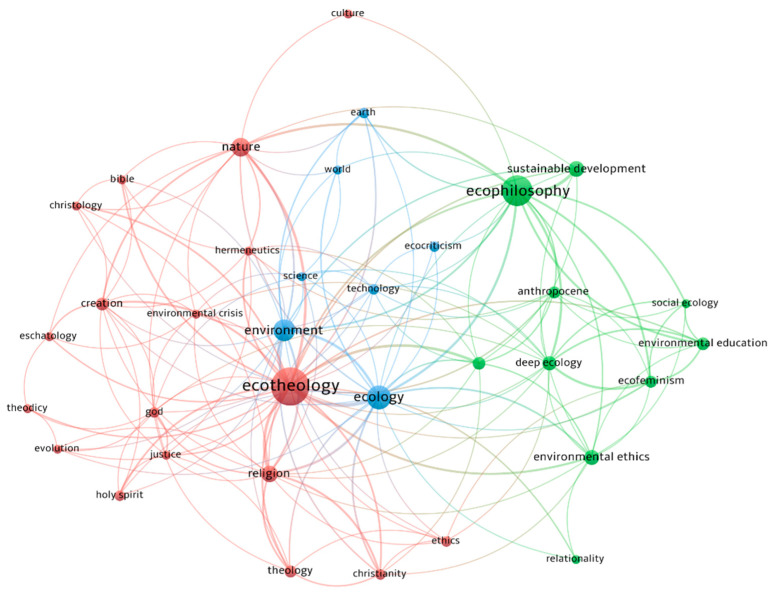
Keywords visualized. Source: Computed in VOSviewer by the author.

**Figure 5 ijerph-18-06572-f005:**
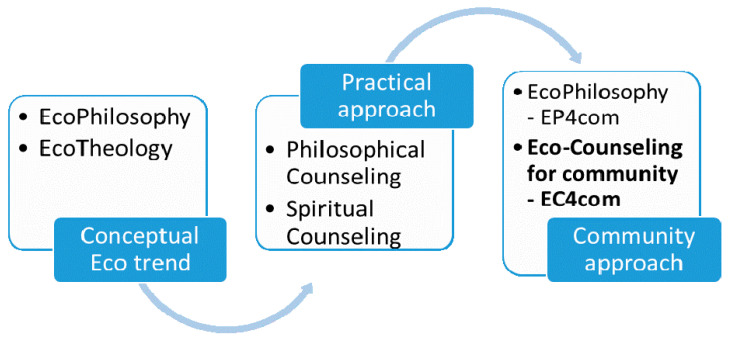
Eco-trend and practical counseling for communities.

**Figure 6 ijerph-18-06572-f006:**
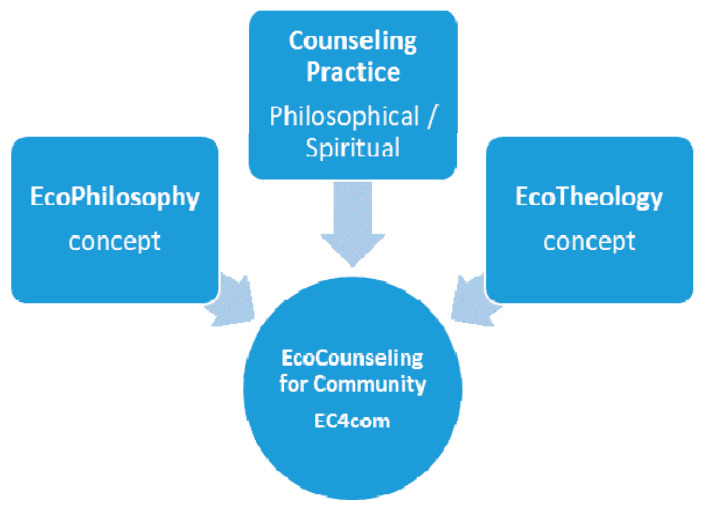
Initiation of the community counseling practice program.

**Figure 7 ijerph-18-06572-f007:**
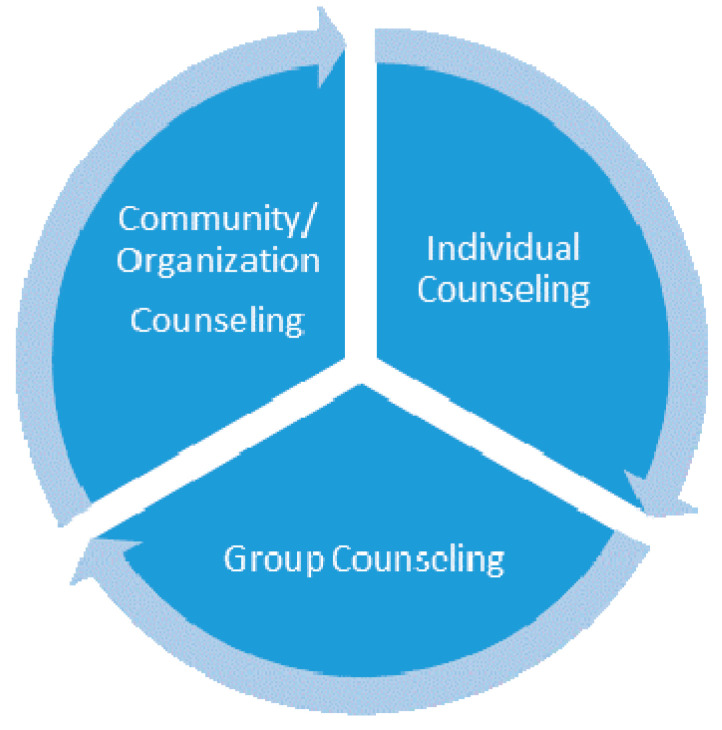
Application topics of community counseling.

**Figure 8 ijerph-18-06572-f008:**
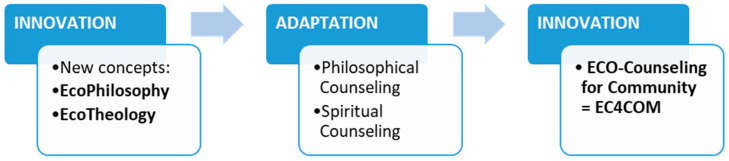
Evolution toward innovation and adaptation of the analyzed fields.

**Table 1 ijerph-18-06572-t001:** Comparative elements of the spiritual and philosophical counseling.

Comparative Elements/Features	Spiritual Counseling	Philosophical Counseling
The context of the activity	Belonging to the church/cult	Without determined context
The purpose of the action	Permanent active service, parish type, with missionary vocation	Practices based on instruments of philosophy and transferred by philosophers to counseling
The destination of the action	Christian community	Customers individuals or groups of people
The object of the action	The person’s relationship with the divinity. It produces changes in counselee thinking	The life examined/supports the clarification of the vision about the world and the person’s life
Main way of working	It is based on listening/dialogue	Through individual/collective dialogue
The professionalism	No deontological norms	A code of ethics is applied
The trend of professionalization	Remains attached to the church/parish	It can be regulated as a specialized public service
Operator training	The specialist comes from the parish, has theological studies no continuous training required	Philosophers who become practitioners or specialists trained in philosophical counseling, they need to follow continuous training
Freedom of conscience of the operator	In agreement with love for/relationship with the divinity (agape)	It derives from the worldview of the philosopher/client
Organizational destination	Less/Can be applied to groups of parishioner	Applies to organizations/institutions/communities
The meaning of the action	What does the world of the believer looks like?	Analysis of life in all its forms
Listening of the persons by the operator	With empathy/warm/encouraging. Try to reveal the person’s past in his approach	It is included in the dialogue of the parties, a compassion is manifested, in the form of a friendship of intellectual nature
How one can activate the person’s resources	By stigmatizing negative actions	By stimulating positive actions
Operating requirements	Attached to the church/parish, using the existing material base	Requires a location/office set up for counseling process
Type of the service provided	Free of charge	Onerous, the fee payment

Source: The information is summarized by the author from reference [78].

**Table 2 ijerph-18-06572-t002:** Clusters.

Cluster 1 (Red)	Cluster 2 (Blue)	Cluster 3 (Green)
Ecotheology	Ecophilosophy	Ecology
Nature	Sustainable development	Environment
Religion	Deep ecology	Earth
Creation	Climate change	Eco-criticism
Theology	Environmental ethics	Technology

**Table 3 ijerph-18-06572-t003:** Top authors by citations.

Author	Documents	Citations	Average Citations	Topic
Latour, B.	1	45	4.02	Ecotheology
Harrison, P.	1	41	2.28	Religion
Whitney, E.	2	37	3.01	Ecotheology
Fairweather, P.G.	1	33	1.47	Ecophilosophy
Francis, G.	1	25	1.11	Ecology
Hull, Z.	2	24	5.45	Sustainable development
Rozzi, R.	1	24	7.38	Ecology
Riley, M. T.	3	17	9.62	Ecology
Wersal, I.	1	16	2.09	Environmental ethics
Fox, W.	1	13	3.25	Ecophilosophy
Bratton, S.	2	11	2.11	Ecotheology
Skrimshire, S.	1	11	2.40	Climate change
Booth, A.	2	10	2.44	Environmental spirituality
Piatek, Z.	3	10	5.14	Ecology

## Data Availability

The data were collected from the following link: https://appswebofknowledge-com.am.e-nformation.ro/WOS_GeneralSearch_input.do?product=WOS&search_mode=GeneralSearch&SID=F3nCA7R1IK5L87kILvR&preferencesSaved=.

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
