# Peer review of "Eco Trends, Counseling and Applied Ecology in Community Using Sophia"

_ijerph, 2021, doi:10.3390/ijerph18126572_

Round 1

Reviewer 1 Report

The paper is peppered with the word "specific" without giving many specifics. It attempts to straddle between neutral scientific analysis and eco-advocacy. 

It basically says that sophia broadly defined is gaining currency based on keyword searches. 

There may be an audience for this paper but I do not find much of interest in it. 

But then I don't find much of interest in the assumption that people are going to wake up, get philosophical and compromise their advantages to bring about a better society. I do not see much evidence of that and hence am more interested in more pragmatic, less abstracted approaches to bringing about our much-needed ecological change. 

Author Response

Thank you for reviewing the paper and the recommendations made to improve it, which were implemented as follows:

The paper is peppered with the word "specific" without giving many specifics. It attempts to straddle between neutral scientific analysis and eco-advocacy. It basically says that sophia broadly defined is gaining currency based on keyword searches. There may be an audience for this paper but I do not find much of interest in it. But then I don't find much of interest in the assumption that people are going to wake up, get philosophical and compromise their advantages to bring about a better society. I do not see much evidence of that and hence am more interested in more pragmatic, less abstracted approaches to bringing about our much-needed ecological change.

Thank you for your patience and attention in analyzing and reviewing our work. We paid close attention to each comment and suggestion received, considering other opinions expressed in the review process, meaning that we made the necessary corrections or additions to improve it, in order to publish the paper in the special edition Review in Environment and Applied Ecology of the IJERPH journal.

Indeed, I used the word ‘specific’ in the translation of the paper in excess and I made all the necessary corrections in the text. Thank you for reporting the error, it was removed.

The ecological field also implies more pragmatic approaches, as there is research in the field of science, but I also found the existence of such an interest in research in the social sciences and humanities and precisely for this reason I wrote this paper.

In order to achieve a 'awakening' of humanity with the help of Sofia, I propose in the paper (in the discussion section and in conclusions) the activation of practices that derive from the two analyzed fields, and that can successfully take over the ecological trend that we support to transform it into practices destined for the community, through specialized counseling practices, which can also take over the “eco” analyzed concepts and bring them for the benefit of the community and implicitly of the people within them.

I appreciate the opinions expressed for the review of the paper, hoping that by publishing the paper we will open the way for other approaches or future research of the analyzed fields, for the benefit of humanity and the planet, by involve in the process as many knowledge fields.

Reviewer 2 Report

The article is interesting by comparing philosophical counseling and spiritual practices. Sometimes the author admits the idea that they are practically the same thing. But the comparative table shows the essential difference in method and content. I consider the integral direction EC4com justified by the author to be significant. The author's thorough work with articles from the WoS list should be noted. The factual content of the article is at a very high level, competent statistics, visual schemes.

additional comments:

Indeed, I think that some points could be clarified:
First, by what criteria does the author establish the distinction and similarity between philosophical and pastoral counseling? It is probably better to describe these criteria more clearly.
Second, the author notes that philosophical and spiritual counseling adapts their practices to the needs of modern society. It is necessary to give a description of those modern needs (the most urgent), which these practices correspond to.
Third, it would be useful to show some techniques for what the author calls "Eco-Counseling for community".

Author Response

Thank you for reviewing our paper and expressing your views for publication in the special edition entitled: Review in Environment and Applied Ecology of the journal IFERPH, and I hope that the topic will generate new approaches to the areas analyzed, opening new directions, both for research and in the direction of applying eco trends in practices (including counseling) for contemporary society.

The recommendations made to improve it were implemented as follows:

The article is interesting by comparing philosophical counseling and spiritual practices. Sometimes the author admits the idea that they are practically the same thing. But the comparative table shows the essential difference in method and content. I consider the integral direction EC4com justified by the author to be significant. The author's thorough work with articles from the WoS list should be noted. The factual content of the article is at a very high level, competent statistics, visual schemes. Indeed, I think that some points could be clarified: First, by what criteria does the author establish the distinction and similarity between philosophical and pastoral counseling? It is probably better to describe these criteria more clearly.

For a better clarity of the paper, we decided to modify table no. 1 and the restructured presentation of the information, in order to highlight the criteria and elements of the comparative analysis used for the two practices, to identify the differences, but also some similarities, aspects that were commented in the text, according to the new table inserted in the paper.

Second, the author notes that philosophical and spiritual counseling adapts their practices to the needs of modern society. It is necessary to give a description of those modern needs (the most urgent), which these practices correspond to.

In additions were made in the discussion section regarding the request for this clarification.

Third, it would be useful to show some techniques for what the author calls "Eco-Counseling for community".

The paper (discussion section) has been revised and supplemented with information on the techniques that can be used in the practices proposed in the paper.

Thank you once again for reviewing the paper and for the clarification suggestions submitted, which have been included in the revised version of the paper for publication.

Reviewer 3 Report

Thanks for providing me the opportunity of reading and commenting your paper. 

At first I believe that the document would gain value if a carefull proof is made to avoid typos (such as “ecophilospohy", innovation appearing with a single n  - figure 8.....) spelling mistakes and misleading sentences. These flaws do not meet the standards of the Journal.

The concepts presented are new/ heterodox however, I believe that there is no underlying scientific reasoning for the cnnection between the concepts and counseling and its efficacy in terms of community adoption.

Moreover, counseling appears in different levels and perspectives, which were not connected to the previous literature.

Additionally, the theoretical aspects are not connected. Using the example of table 1. Where do these aspects come from? The elements were mentioned, but no literature supports their differences or similarities. Why were these points raised? How can we know that there isn't more? 

Then, the differences and the similarities are not clustered to draw a profile.

Why is the 10 year trend evidenced when there is no defined trend? The concept is volatile over time, and appears as declining over the last two years. Was it expectable?

Figure 4 is not explored. The lack of explanation about the keyword connection makes us pose the question why is this here?

What is the result of presenting authors per country as well as citations if no general trend is designed.

Are the research questions answered by the biblimetric results? In what extent? Is is proved by the literature?

What are the theoretical and practical implications of your research? Can you effectively draw a policy recommendation based on you findings? Which one? Why? 

The conclusions are vague, and do not draw a clear map for the future: The analysis performed is comparative one of the specific characteristics: "resulting some similarities and essential differences, which individualize the counseling, and are based on the tools, concepts and methods specific to each area" 

What are the expected outcomes of your framework?

What is the societal gain? 

I believe that it is a good perspective, however the research, as is, is very embryonic and does not shed light on the importance of these practives to leverage the adoption of eco-innovative practices through counseling. 

I wish the authors all the luck for their research.

Author Response

Thank you for reviewing the paper and for the valuable suggestions for improving the paper, which were introduced in its content in order to clarify and better present the topic.

The recommendations made to improve it were implemented as follows:

Thanks for providing me the opportunity of reading and commenting your paper. At first I believe that the document would gain value if a carefull proof is made to avoid typos (such as “ecophilospohy", innovation appearing with a single n  - figure 8.....) spelling mistakes and misleading sentences. These flaws do not meet the standards of the Journal.

Thanks for reporting the drafting errors, the text has been revised from this point of view and the grammatical expressions (including the correction of figure 8) and at the end the paper was proofreading checked.

The concepts presented are new/ heterodox however, I believe that there is no underlying scientific reasoning for the connection between the concepts and counseling and its efficacy in terms of community adoption. Moreover, counseling appears in different levels and perspectives, which were not connected to the previous literature.

The paper refers to the theoretical concepts identified in the theory of the two areas analyzed, after which they are studied in terms of their potential to be assimilated and included in the practices of the two areas, manifested in the form of counseling, philosophical and spiritual. These practices are relatively recently activated, initially at the level of individuals and groups, and the proposal of the paper in the discussion and conclusion sections refers to their application in the wider context of a community that needs the ecological trend of the concepts identified in the theoretical section of each domain analyzed.

These are the arguments for which the theoretical section dealt mostly with the hybrid concepts ecophilosophy and ecotheology, precisely to highlight the potential of this conceptual trend to be implemented in the practices developed by each of the areas analyzed, and the whole process is schematically presented in the figures 5-6.

Additionally, the theoretical aspects are not connected. Using the example of table 1. Where do these aspects come from? The elements were mentioned, but no literature supports their differences or similarities. Why were these points raised? How can we know that there isn't more? Then, the differences and the similarities are not clustered to draw a profile.

In order to eliminate any confusion on the comparisons initially presented in table 1, it is restructured in a new format (the material being an adaptation of the authors after a studied reference), this presenting the characteristics analyzed in the comparative table, in order to show the differences or similarities, which emphasize the originality of each practice, their origin and adaptation to the practice of counseling in the field from which it comes. The table presented has his limits, through the elements taken in the analysis, but we opted for this presentation of some features to open other research opportunities, for those interested in developing the topic addressed in the paper.

Why is the 10 year trend evidenced when there is no defined trend? The concept is volatile over time, and appears as declining over the last two years. Was it expectable?

We stopped our presentation the last 10 to reflect them in fig. 3 since this period turned out to be a representative one, with obvious tendencies of interest on the analyzed field and confirmed by the evolution with increasing trend of the number of papers published about the analyzed fields in the paper.

Figure 4 is not explored. The lack of explanation about the keyword connection makes us pose the question why is this here?

Thank you for this reporting to complete the explanations on the figure used in the analysis, this being now explained in the content of the paper, in accordance with the suggestions received.

What is the result of presenting authors per country as well as citations if no general trend is designed.

Thank you for this report that was completed in the revised paper by additional details in the bibliometric analysis performed in the paper.

Are the research questions answered by the biblimetric results? In what extent? Is is proved by the literature?

We used bibliometric analysis in order to find out what were the trends of those who wrote about these fields, to identify the clusters formed and the links between the concepts presented in the papers published for the period under review. The research questions are confirmed by the results of the investigation, which encourages the involvement of the ecological trend in the practices developed in each field, and which can find new destinations, one being the community, for which we propose the EC4com concept.

What are the theoretical and practical implications of your research? Can you effectively draw a policy recommendation based on you findings? Which one? Why? The conclusions are vague, and do not draw a clear map for the future: The analysis performed is comparative one of the specific characteristics: "resulting some similarities and essential differences, which individualize the counseling, and are based on the tools, concepts and methods specific to each area". What are the expected outcomes of your framework? What is the societal gain?

The Results and Conclusions sections were completed accordingly, in order to highlight the implications of the topic studied in the paper. We consider the proposal we come up with an innovative idea, which will produce effects by applying it within the communities, which can be recipients of applied counseling services, which can take over the ecological trend manifested at a conceptual level. It remains as a future direction of study, which of validating the effect it had on society, through the ongoing evaluations that can be made within the communities that will take the initiative proposed in the paper, respectively they will apply EC4com. Counseling services can be successfully included in community development policies and strategies, given that the proposed specialists and practitioners can become important vectors of communication, both by their position within the community and by the nature of the services offered to the community.

The paper generates only one direction, given by the innovation brought by the proposed EC4com concept, which can be initially implemented among the practitioners we referred to, and will be promoted and evaluated in the community, for which it is intended, and its evaluation will be made during its implementation.

I believe that it is a good perspective, however the research, as is, is very embryonic and does not shed light on the importance of these practives to leverage the adoption of eco-innovative practices through counseling. I wish the authors all the luck for their research.

Thank you for your patience and attention in analyzing and reviewing our paper. We assure you that we have paid close attention to every comment and suggestion received, and we have taken into account all the opinions expressed in the review process, meaning that we have made the corrections or additions suggested by all reviewers, in order to improve the paper and publication in the special edition: Review in Environment and Applied Ecology of the journal.

We appreciate the recommendations for improving the material and thank you for reviewing the paper in order to publish it in a Special Issue dedicated to this topic, which paves the way for other research in the field, and through the IJERPH journal.

Round 2

Reviewer 3 Report

Many thanks for providing a new version of the article. 

As I can see none of my recommendations were taken into consideration and the new version is very similar from the first. 

"specific" was removed, 2 references were added and table 1 was formated.

I feel that the authors should consider my review, however I respect them for not following the recommendations provided. 

I wish the research the best of luck.

Author Response

Thank you for your opinion, which unfortunately wasn't favorable, and I confirm that the paper was reviewed and first response was based on the suggestions received, containing specific references to the corrections and additions made, and arguments to support the paper. The corrections regarding the excessive use of the word ‘specific’ were made at the suggestion of another reviewer, which proves that we took into account all the reported suggestions, including those favorable to the current format of the paper. In order to highlight all the corrections made, they were marked in the final text using the color blue (additions / completions of some expressions, or the addition of new paragraphs) together with table 2 which was replaced with the current form, in order to respond to the review. I also marked in the text using red color the words or expressions removed, as well as the initial table marked with no. 2. I assure you that I took very seriously the review received , and the corrections made prove the good faith to make a good presentation of the topic and to be in accordance with all the suggestions received from reviewers, but also taking into account the purpose proposed by this research. I think about the paper can open new research projects, which will evaluate the implementation of our proposal in community. For the additional argumentation, in the final version of the Discussions section, I have added the following comments, which I hope will be able to clarify and support the paper to publish it.

The approach of the topic was made from the position of a research on the practices of counseling, and choosing the spiritual and philosophical counseling, was made to identify the conceptual links in the basic fields from which it derives, respectively theology and philosophy, to study the options to interfere with other field, that of ecology and environmental protection. The purpose of this approach is to present the way of adapting the respective practices to the counseling applied to the person, with the identification of a new trend in contemporary society. The paper uses a comparative analysis of the main characteristics identified for the mentioned practices, respectively spiritual counseling and philosophical counseling, in order to identify differences or common approaches, to find their potential for adaptation and innovation, by attracting new concepts, derived from the eco trend presented. Starting from the presented concepts, we also included their bibliometric analysis, the results being interpreted to show the existing connections between the basic domains, as well as the relatively low publishing options manifested for the analyzed domains. All these studies and comparative analyzes have led us to the idea of ​​developing the analyzed practices, which may include the ecological trend in a process of specialization of practitioners, a trend that can be considered the premise of a process of adaptation to contemporary society, through Eco Counseling, the practice proposed in the paper. Returning to the practice of counseling, were presented in the paper (fig.7) who are the beneficiaries of this service, from which we can expand its scope, by promoting Eco-Counseling in communities, a process that involves the option of applying it to people, groups or other entities from community. We consider the approached topic just a preliminary one, which can open new directions of approach and research of the concepts and practices analyzed now, by developing a program that can be implemented in the process of training and continuous training of counseling practitioners. Thus, the specialization proposed by Eco Counseling for community (EC4com) we believe can become a useful program for communities, which can be monitored during its implementation and evaluated at the end, in order to improve it, or simply to report the effects and benefits resulting from this new practice.

I assure you of the same consideration that you showed in this review and I support again the revised final form of the paper, according to attachment version (ijerph-1246670.rev2).

Round 3

Reviewer 3 Report

Onve more, thanks for the opportunity of reading your work and raising some comments in regards to its evolution. 

At first I would like to ask the authors for reading my former comments, as it is very disappointing that they were not taken into consideration in the paper.

Onve more, I beleive that the paper should go through a deep proof reading to improve the language and increase the readability. Also, the para graphs are too long and make the understanding of the content difficult.

Section 2, as I have already mentioned, does not connect the concepts building a solid argument. 

Section 3 explores two concepts, however there is no link among them, moreover, what is the underlying reason for connecting the concep to ecology? what is the expected outcome?

Also, section 4  presents the papers and the authors, however no link is made with the extant research or even drawn new avenues of research.

Naturally, the conclusions section will benefit from the identification of the trends as well as the gaps. 

All the best with your research.

Author Response

We are at an impasse with the revision of the paper, in the sense that you say that we don't take into account your suggestions. We carefully reviewed all the answers we formulated but in the first round we already provided our answers and arguments on the content and form of the paper and referred to each of the corrections made in the text (suggestively marked in color ) based on all suggestions received from reviewers.

In the second round we have again the situation in which the corrections made and the arguments presented are considered non-existent or minor, reason for which we included in the paper in the discussion section a detailed paragraph in which we argued the revised format of the papers, and appreciating the fact that you can be agree with the decision of the authors not to follow all the recommendations received.

In the third round we find that this statement is no longer valid, asking more arguments and changes to the paper.

Only in the third round was it found that the paper needs the proofreading process, although we know it can be verified in this regard and through MDPI support, before publication.

The paragraphs have the respective form to give the meaning of an idea, and I think a separation of them can affect the content of the paper, because the approaches related to philosophy cannot be written in short sentences.

The theoretical section brings to attention the concepts that come together, which do not require a different approach, the paper going on the idea of creating and supporting a common program, which brings together in practice all the concepts presented.

Section 3 of the paper refers strictly to the research methodology without requiring the additional arguments.

In the results section, in addition to the comparative analysis of the practices proposed for the final conclusion, we have the bibliometric analysis of the theoretical concepts, to highlight the connections that exist and justify the proposal for their collaboration for community.